# AutoCoG: A Unified Data-Model Co-Search Framework for Graph Neural Networks

Duc Hoang[1]  Kaixiong Zhou[2]  Tianlong Chen[1]  Xia Hu[2]  Zhangyang Wang[1]

[1]University of Texas at Austin
[2]Rice University

**Abstract**     Neural architecture search (NAS) has demonstrated success in discovering promising architectures for vision or language modeling tasks, and it has recently been introduced to searching for graph neural networks (GNNs) as well. Despite the preliminary success, GNNs struggle in dealing with heterophily or low-homophily graphs where connected nodes may have different class labels and dissimilar features. To this end, we propose co-optimizing both the input graph topology and the model's architecture topology simultaneously. That yields **AutoCoG**, the first unified data-model co-search NAS framework for GNNs. By defining a highly flexible data-model co-search space, **AutoCoG** is gracefully formulated as a principled bi-level optimization that can be end-to-end solved by the differentiable search methods. Experiments show that **AutoCoG** achieves an average performance gain across all datasets of 3.18% over the following best approach, and ranks best against all other state-of-the-art methods with an average ranking of 2.5. Code is available at `https://github.com/VITA-Group/AutoCoG`.

## 1 Introduction

Graph neural networks (GNNs) have emerged as promising tools to analyze networked data in various real-world scenarios, such as social media (Grover and Leskovec, 2016) and biochemical graph analytics (Zitnik and Leskovec, 2017). Specifically, GNNs apply recursive message passing to learn the embedding representation of each node via aggregating the representations of its neighbors and itself. Motivated by the significant success of node embedding learning, plenty of GNN variants have been explored for the diverse downstream graph analysis tasks, including GCN (Kipf and Welling, 2016a), GraphSAGE (Hamilton et al., 2018), and GCNII (Chen et al., 2020a).

However, training GNNs is notoriously challenging, more so when they are trained under heterophily or disassociative graphs, not to mention deep GNNs (Chen et al., 2020a; Zhou et al., 2020). First, since graphs abstract diverse data sources and present tremendous heterogeneity, the success of GNNs is often accompanied by extensive tuning of model architectural hyperparameters to characterize specific graph data. For example, it was reported that graph attention networks (GAT) (Veličković et al., 2018) are sensitive to the number of attention heads, which has to be carefully searched for the citation networks and the protein-protein interaction data, respectively. Second, in the real world, graphs often opposites attract which inevitably lead to noisy setting, where GNNs tend to suffer from overfitting and generalize poorly to the unseen testing data. Third, despite the potential of deep GNNs in learning the informative high-order neighborhood, the training of deep GNNs is widely known to be limited by the issues of over-smoothing, gradient vanishing, and over-squashing (Chen et al., 2020a).

Recently, the automated graph neural architecture search (NAS), graph augmentation tricks, and deeper architectures have been independently proposed to tackle the above GNN training challenges partially. Expressly, most of the existing automated efforts are limited to neural architecture tuning, while graph augmentation is often overlooked despite often being effective to gain performance (Li and King, 2020; Zhou et al., 2019a). This is primarily because changes to

the existing graph structure could have a cascading effect on the process of information aggregation, but adds a new layer of complexity to the above complex architecture tuning problem. Additionally, existing GNN NAS works are known to scale poorly in deeper architectures. This is primarily due to the exploding search space which makes searching unstable. Previous efforts have limited themselves in searching the shallow GNNs with less than 3 layers. Finally, Figure(1) illustrates circumstances where the aggregation mechanism fails due to unfavorable graph topology thus, it remains a daunting task to optimize the design philosophy for GNNs comprehensively.

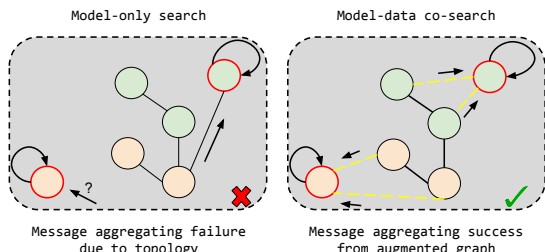

To bridge the gaps, we propose **AutoCoG**, the first NAS framework towards **unified data-model co-search for GNNs** to specifically tame the problem training under heterophily condition. Besides automatically optimizing the GNNs' neural architecture, we propose to simultaneously optimize the input graph topology, via progressively growing and pruning using a separate GNN model to learn to attune the graph to the proposed architecture. Additionally, by defining the highly flexible data-model co-search space, **AutoCoG** is formulated as a principled bi-level optimization that can be end-to-end solved by the differentiable search methods. To scale up our core framework to searching deep GNN architectures, we curb an

Figure 1: In red outline are nodes with poor aggregation, black arrows, due to a graph topology under heterophily. This can be mitigated by learning to place meaningful edges, yellow lines, to facilitate proper message propagation. Motivate to solve this performance problem, we propose co-adapting both graph and model in a end-to-end manner.

explosive search space as the number of layers increased by performing multiple searching stages with increasing depth, as inspired by Chen et al. (2019c). Additionally, for each search stage, we evolve the graph by growing/pruning it at the same time. To stabilize the search landscape from the shifting topologies of graph and model, we further utilize (Chen et al., 2020a) to combat the over-smoothing/over-squashing issues.

Together, our framework ensures a reliable way to discover the powerful architectures, a stable model training environment, and state-of-the-art results to train graphs of different degrees of homophily. **AutoCoG** searches for and trains on deep or shallow graph neural networks to successfully deliver superior results in Web datasets, Actor, Coauthor, and Wikipedia benchmarks. In summary, our three contributing novelties are:

- We propose **AutoCoG**, the first NAS framework towards unified data-model co-search for GNNs. Our bi-level optimization formulation uniquely enables the end-to-end discovery of GNNs' neural architecture and graph topology altogether.

- We perform extensive analysis the resulting learned graph-structures for each benchmarks. To strengthen the co-search framework, we organically integrate several techniques to directly combat issues of searching unreliability, training instability, and scalability, that have previously plagued NAS approaches for searching deeper GNNs.

- Experiments show that **AutoCoG** achieves an average performance gain across all datasets of 3.18% over the following best approach, and ranks best against all other state-of-the-art methods with an average ranking of 2.5.

## 2 Related works

**Graph neural networks**. Motivated by the state-of-the-art results of GNNs in graph analytics, there have been numerous GNN variants (Bruna et al., 2013; Hamilton et al., 2017; Xu et al., 2019;

Chen et al., 2020a; Wu et al., 2019a; Zhou et al., 2021b). Most of these existing approaches fit within the category of spatial GNNs. Namely, following the spatial message passing strategy, the core idea of GNNs is to learn the embedding representation of a node by aggregating the embeddings of its neighbors and node itself recursively. The previous empirical studies show that GNNs often achieve the best performance with less than 3 layers (Kipf and Welling, 2016a; Veličković et al., 2018). Key limitations of GNNs are their performances decrease significantly with the increasing of model depth and the degree of graph homophily they operate on. As the graph convolutional layer increases, the node representations will converge to indistinguishable vectors due to the recursive neighborhood aggregation and non-linear activation (Li et al., 2018; Oono and Suzuki, 2020; Zhou et al., 2021a; Guo et al., 2021), which is well known as the over-smoothing issue (NT and Maehara, 2019; Chen et al., 2019a; Alon and Yahav, 2020; Chien et al., 2021; Huang et al., 2020).

**Graph augmentation**. Data augmentation methods has been widely applied to improve the generalization performances of deep neural networks, such as convolutional and recurrent neural networks (Shorten and Khoshgoftaar, 2019; Antoniou et al., 2017; Feng et al., 2021). They aim to craft the out-of-distribution training data to avoid overfitting with the customized augmentation policies. In the graph analytics, GNNs are prone to overfit the naturally noisy training graphs, which may miss the ground-truth nodes/edges or contain the erroneous information (Zügner et al., 2018). Different from the grid-like image data, the graph augmentation is often operated on the adjacency structure or node features (You et al., 2020b, 2021, 2022; Lai et al., 2020). Existing graph augmentations could be categorized into the following two classes. (i) The random augmentation either drops/adds edges to modify the graph, or masks parts of the node features (Rong et al., 2020a; You et al., 2020b; Feng et al., 2020). (ii) The differentiable augmentation learns to optimize the adjacency affinity matrix by minimizing the concerned task loss. Based upon the computed affinity matrix, the differentiable augmentation either continuously combines it into the original adjacency matrix (Zhao et al., 2020b; Chen et al., 2020b), or samples the discrete edges to formulate new graph (Chen et al., 2019b).

**Neural architecture search**. Targeting at alleviating the laborious hyperparameter tuning, NAS automates the designing of good neural architectures for any a given application. It is shockingly reported that the searched neural architectures could outperform the human-designed ones in many real-world scenarios, such as image classification (Zoph and Le, 2016; Zoph et al., 2018) and generation (Wang and Huan, 2019; Gong et al., 2019). Most of NAS frameworks apply one of the following search algorithms: reinforcement learning (RL) (Pham et al., 2018; Baker et al., 2016), evolution algorithm (EA) (Liu et al., 2017; Miikkulainen et al., 2019; Xie and Yuille, 2017), and one-shot differentiable search (Liu et al., 2018; Zela et al., 2020). There are several recent efforts to conjoin the researches of GNNs and NAS (Gao et al., 2019; Zhou et al., 2019a; You et al., 2020a; Ding et al., 2020; Zhao et al., 2020a). However, all of them are limited in exploring the shallow GNNs, and fail to denoise the underlying graph to further ameliorate the model performance. In this work, we aim to simultaneously search the deep GNN models and graph structure to optimize the downstream graph analytics.

**Co-adaptive search between graph's structure and model's architecture**. GASSO (Qin et al., 2021) is a recent work that similarly proposes the idea of model-graph co-search. Yet two differentiation factors uniquely defined our two works. Firstly, GASSO is a technique that learns attention coefficients G∈[0, 1] only for existing edges E, which is mathematically equivalent to graph attention neural networks (Qin et al., 2021). In contrast, **Auto-CoG** directly modifies the graph's structure via pruning poorly attended edges and adding new un-seen edges. Thus the derived graph structure is unique from the original underlying graph. Finally, GASSO employs a coarse macro-level search space with only eight operators and two layers. Its design decision space is shallow (small),

consisting of only 256 unique combinations. Theoretically, by searching the optimal attention function in our **Auto-Cog**, we could approximate the "attentional structure learning" in GASSO.

## 3 Methodology

**Preliminary**. We briefly review the basic of the definition of homophily and the message-passing based graph convolution networks (GCN). The homophily or edge-homophily ratio of a graph measures the ratio between intra-node pairs $(i, j)$ overall all edges $\mathcal{E}$ and is given as:

$$\frac{|\{(i, j) : (i, j) \in \mathcal{E} \wedge y_v = y_w\}|}{|\mathcal{E}|}, \tag{1}$$

where $\mathcal{E}$ denotes the set of edges in the graph, and $|\mathcal{E}|$ denotes the cardinality of edges. Being defined as the message passing along the edges of graph, the $k$-th layer of GNNs could be generally written as:

$$\begin{aligned} h_i^{(k)} &= \text{AGGR}(\{a_{ij}^{(k)} W^{(k)} x_j^{(k-1)} : j \in \mathcal{N}(i)\}), \\ x_i^{(k)} &= \sigma(\text{COMB}(W^{(k)} x_i^{(k-1)}, h_i^{(k)})). \end{aligned} \tag{2}$$

$x_i^{(k)}$ denotes the node embedding of node $i$ at the $k$-th layer. $W^{(k)} \in \mathbb{R}^{D \times D}$ represents the learnable layer-wise weights shared by all the nodes, where $D$ denotes the dimension of hidden units. $a_{ij}^{(k)}$ dictates the attention coefficient between nodes $i$ and $j$ derived from some Attention functions. $\mathcal{N}(i)$ denotes the set of neighboring nodes of node $i$. $h_i^{(k)}$ is the resulting embeddings after applying an AGGR function to aggregate the set of neighboring embeddings from the previous layer. In addition, function COMB incorporates information from the node itself with its neighboring embeddings $h_i^{(k)}$, and $\sigma$ provides the nonlinear activation.

### 3.1 Unified Data-Model Co-Search Space

**3.1.1 Model Search Space**. The design of model search space should achieve a balanced trade-off between the diversity and efficiency Zhou et al. (2022). Although a large search space subsumes the diverse GNN architectures to adapt to the different graph analysis tasks, it would be extremely time-consuming to explore the optimal design. In the existing search spaces of GNNs (Gao et al., 2019; Zhou et al., 2019b; You et al., 2020a), they often contain the architecture components of hidden units, attention, aggregation, combination, and activation functions, as well as the skip connections. To efficiently

Table 1: The set of attention functions Gao et al. (2019), where || denotes the concatenation operation, $\bar{a}, \bar{a}_i, \bar{a}_j$ denote learnable vectors, $W_G$ denotes the trainable matrix.

| Attention Choice | Expression Form |
|---|---|
| GCN | $\frac{1}{\sqrt{|\mathcal{N}(i)||\mathcal{N}(j)|}}$ |
| COS | $\bar{a}(W^{(k)} x_i^{(k-1)} || W^{(k)} x_j^{(k-1)})$ |
| LINEAR | $\tanh(\bar{a}_l W^{(k)} x_i^{(k-1)} || W^{(k)} x_j^{(k-1)})$ |
| GERE-LINEAR | $W_G \tanh(W^{(k)} x_i^{(k-1)} + W^{(k)} x_i^{(k-1)})$ |
| GAT | $\text{LeakyReLU}(\bar{a}(W^{(k)} x_i^{(k-1)} || W^{(k)} x_j^{(k-1)}))$ |
| GAT-SUM | $a_{ij}^{(k)} + a_{ij}^{(k)}$ based on GAT |
| CONST | $1$ |

search the outperforming shallow and deep GNNs, we compare the effectiveness of each component, and greatly shrink down the search space to focus on three key components: the activation function, the attention module, and the skip connections. They are generally believed to impact GNN's expressive capability and depth scalability (Chen et al., 2021b). We fix the aggregation function and combination function to be simple summation, and treat the hidden units as hyperparameter. Below we lay out our searchable design for them one-by-one:

- *Attention search space*: Attention mechanism has been shown by (Veličković et al., 2018) to effectively stabilize training by placing proper neighborhood scaling with attention coefficient $a_{ij}$. We list our attention choices in Table 1.

- *Activation search space*: For the activation functions, we search among these operations: {ReLU, Sigmoid, Tanh, Linear, SoftPlus, LeakyReLU, ReLU6, ELU}.

- *Skip connection search space*: For a $L$-Layer GNN, various skip connections can be applied to overcome the effect of over-smoothing. Previous deep GNN works (Chen et al., 2020a; Zhang et al., 2020; Chen et al., 2021b) illustrate a significant correlation between the type of skip connections and the model performance. We include two categories of skip connections: (i) Initial Connection, and (ii) Jumping-Knowledge aggregation.

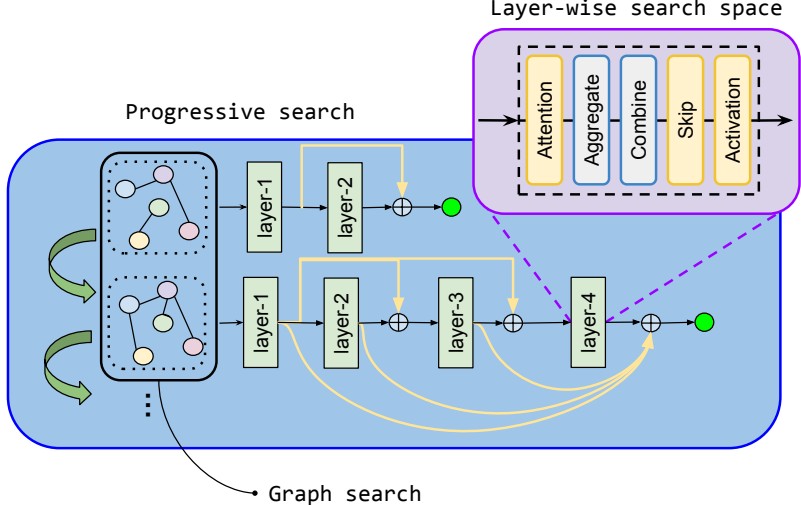

Figure 2: An illustration of **AutoCoG** framework. We marked the components we searched on as yellow; this notation also extends to the different skip connections illustrated in the progressive search box. Furthermore, we narrow down our search choice for each step within progressive search while extending the model's layers. We also perform graph augmentation for every step. Further details on the process can be found in Algorithm(1) in the appendix

**3.1.2 Graph Augmentation**. We often expect a clustering of liked nodes when operating on graph data in a "like attracts like" world. However, in reality, when modeling complex relationships, we may observe the opposite, where node identities are often best described by contrasting with their neighbors in "different attract" relationships. Under such heterophily circumstance, GNNs' performances degrade (Pei et al., 2020a; Zhu et al., 2020a; Battaglia et al., 2018), which makes sense intuitively, since aggregating the unrelated neighbors can lead to class obscurity (Zhu et al., 2020a), i.e over-smoothing. Thus overcoming heterophily with model architecture alone is difficult and often requires complicated, and exotic works flow (Abu-El-Haija et al., 2019; Pei et al., 2020b; Lim et al., 2021). Lately, a number of works (Srivastava et al., 2014; Zou et al., 2019; Rong et al., 2020b; Chen et al., 2021a; Huang et al., 2021) have found that direct graph augmentations with stochastic policies — drop/add edges — can decelerate both the over-fitting and over-smoothing issues in training deep GNNs. By learning the graph's topology and the model's architecture, we naturally adapt our data structure around the model's strength, and co-optimize data flow around the message passing mechanism.

**The scoring function**. Graph $G(\mathcal{V}, \mathcal{E})$ can be expressed in the form of an adjacent matrix $A \in \mathbb{R}^{|\mathcal{V}| \times |\mathcal{V}|}$, where $\mathcal{V}$ is the set of vertices and $\mathcal{E}$ is the set of edges. We learn an edge score matrix $S \in \mathbb{R}^{|\mathcal{V}| \times |\mathcal{V}|}$ such that we rewrite the aggregation step in Eqn (2) as:

$$H^{(k)} = (S \odot A)X^{(k-1)}W^{(k)}, \tag{3}$$

where $H^{(k)} = \{h_i^{(k)} : 0 \leq i \leq |\mathcal{V}|\}$ and $X^{(k-1)} = \{x_i^{(k-1)} : 0 \leq i \leq |\mathcal{V}|\}$. We formally define $S \in [0, 1]$ as:

$$
\begin{aligned}
S &= \sigma(\text{MLP}(Z_{V_{\text{src}}} || Z_{V_{\text{tgt}}})) \\
Z &= f(X, G(V, \mathcal{E}); W_s)
\end{aligned}
\tag{4}
$$

Where $f(.; W_s)$ is simply the classic VGAE model by (Kipf and Welling, 2016b), $Z_{V_{\text{src}}}/Z_{V_{\text{src}}}$ are the source and target nodes available from $G(V, \mathcal{E})$, $||$ denotes the concatenation function, and $\sigma$ is the sigmoid function. $S$ is therefore a sparse matrix with only $|\mathcal{E}|$ number of scores.

**Edge growing and pruning**. Taking the advantage of Progressive NAS workflow (Chen et al., 2019c), at each searching stage, we prune the bottom $p$-percentile from $S$, and at the same-time we grow our graph by appending $k$ new edges for each node via embedding similarity. This embedding similarity function is defined as:

$$
\text{Sim}(v_i, v_j) = \frac{1 + \text{Cosine}(z_i, z_j)}{\log D_{ij}}
\tag{5}
$$

Where $\text{Cosine}(\cdot)$ is the cosine similarity between two nodes' scoring embeddings, while $D_{ij}$ is the shortest distance between them. We illustrate the process of graph topology modification visually in Figure 2 and in the pseudo code of Algorithm 1 in the appendix.

## 3.2 Optimization Formulation and Algorithm

### 3.2.1 A Principled Bi-Level Optimization Formulation.
For the sake of conciseness, we use $\alpha$ as the *model space architecture parameters*, and denote $\mathcal{L}_{\text{obj}}$ as the objective loss function given $\alpha$. With $\alpha$ defined, we further denote $\hat{W} = W \odot m_\alpha$ as the pruned sub-model from the supernet derived according to $\alpha$ description, where $\hat{W}, m_\alpha \in \mathbb{R}^{L \times D \times D}$. Additionally, we can write our augmented graph $\hat{G}$ as $\hat{A} = A \odot S$, where $S$ is defined as our learned scoring matrix. Let $Z$ represents the node embedding matrix for a hypothetical 2-layer **AutoCoG**:

$$
Z = \text{Softmax}((\hat{A}\sigma(\hat{A}X\hat{W}^{(0)})\hat{W}^{(1)})).
\tag{6}
$$

Thus the objective loss function $\mathcal{L}_{\text{obj}}$ for a transductive semi-supervised node classification tasks is formally denoted as:

$$
\mathcal{L}_{\text{obj}}(\hat{G}, \hat{W}, X, Y) = -\frac{1}{|Y|} \sum_{y_i \in Y} y_i \log(z_i).
\tag{7}
$$

Extending from (Dong and Yang, 2019), we formulate our data-model co-search as a **joint bi-level optimization**, to solve $\alpha, S$ concurrently with weights $W$ and data space parameters:

$$
\begin{aligned}
\min_\alpha \quad & \mathcal{L}_{\text{obj}}^{\text{valid}}(\hat{W}(W, \alpha), \hat{G}(S), X_{\text{valid}}, Y_{\text{valid}}) \\
\text{s.t.} \quad & \hat{W}, \hat{G} = \arg\min_{W,S} \mathcal{L}_{\text{obj}}^{\text{train}}(W, G, \alpha, S, X_{\text{train}}, Y_{\text{train}})
\end{aligned}
\tag{8}
$$

Note that $\alpha$ are optimized using the objective loss function on the validation set, while $W, S$ are optimized under training set. Additionally, $\hat{G}$ consists of modified edges, not-shown explicitly in Equation (8), but is illustrated in our Algorithm 1. We adopt the same hard-*Gumbel-softmax* trick (Jang et al., 2017) to differentially optimize architectural variables during search.

**3.2.2 Scaling And Stabilizing Search.** The bi-level optimization problem (8) can be solved by differential search methods, and we adopt the GDAS approach in (Dong and Yang, 2019) by default. However, when exploring deep GNN architectures and larger graphs, the model/data search spaces grow exponentially with the layer depth/graph size. They can be entangled to cause even more serious scalability challenge. That is further amplified by the training difficulty and instability of deep GNNs (Chen et al., 2021b). Indeed, we observe that naively applying GDAS is prone to over-smoothing and search collapse, only yielding very poor architectures when searching for more than three layers. Besides, it is not common for the derived graph and model to have considerable performance variations across repeated experiments, due the stochastic initialization and training.

**Progressive search space.** We follow the idea proposed by (Chen et al., 2019c) (also illustrated in Algorithm 1), to divide search into $N$ progressive stages, with each consecutive stage having a larger or equal number of layers than those previously. At each stage, we greedily remove the least selected options (by taking the mean of Soft-Max across $L$ layers and removing the option with the smallest value) from the data's $p$ parameters or model space, and pass on the shrunk co-search space to the next stage. Note that we do not shrink the number of augmentation policies.

## 4 Experiments

### 4.1 Experimental Settings.

We first list our shared training hype-parameters for all datasets, and then list each dataset's specific particularity. By default, we employ the Adam-optimizer (Kingma and Ba, 2017) to learn edges' scores, model's architecture and model's weights with equal learning rate of 0.005, and a $L_2$ regularization of 0.0005. We set the hidden-dimension $D$ to be 256 with a dropout rate of 0.6. As for P-DARTS, for every stage we prune the bottom 10% of edges, and add one new edge per node, and the number of stages are set to be 4, starting from 2 layers, and with a 2 layers increment. Furthermore, for Identity-Mapping, $\gamma$ is set to be 0.5. We search/train for 1000 epochs, while setting our rate of patient to be 400 and 200 respectively. To get our final results, we train the network 10 times to get the average and standard deviation.

The only notable exception to the default settings are the Co-author datasets, where we set the dropout rate to be 0.8 and 0 for CS and Physics respectively. We typically only search between two and eight layers. Finally, for all datasets, we average their accuracy over 10 runs, with random seed between 0 to 9.

### 4.2 Ablation Studies

Table 2: Ablation results comparing the test results between different searching modes at increasing degree of homophily with fixed depth of eight. Best results are bold.

| Experiments $\mathcal{H}$ | Actor 0.375 | Texas 0.411 | Wisconsin 0.488 | Cornell 0.567 | CS 0.827 | Photo 0.833 |
|---|---|---|---|---|---|---|
| Co-search | **38.039±0.16** | **78.378±2.21** | **80.392±0.00** | **64.864±2.97** | **91.840±0.60** | **83.204±2.42** |
| Data-only | 23.924±1.86 | 64.324±1.71 | 46.666±2.41 | 46.486±1.13 | 80.225±2.12 | 82.255±2.72 |
| Model-only | 36.394±0.07 | 72.070±0.85 | 70.588±1.60 | 56.216±1.14 | 88.599±0.86 | 62.798±5.51 |

**Model-graph codependancy.** We justify the need for model-graph co-search by performing three experiments, namely — co-search, data-search, and model-search — to illustrate the respective effectiveness of the individual components which constitutes our framework. We collected these results from several datasets at a fixed depth of 8 while maintaining identical searching settings for all experiments. Note that for data-search, we substitute our model with the vanilla GCN (Kipf and Welling, 2016a). The results are collected in Table(2). Herein our results speak for themselves; we observe a significant improvement in performance, especially for graphs under heterophily, utilizing co-search over model-only and data-only search.

**Correlation between depth and performance**. We observe that an increasing depth does not always positively correlate to performance gain. Homophily negatively correlates to our performance at depth. To explain this phenomenon, we offer this hypothesis: since the number of layers in a model correlates to the number of k-hop neighbors observed, graphs under heterophily need to observe a much larger sub-graph to aggregate meaning information against the inherent noisy neighbors.

In contrast, with an increasing degree of homophily, more layers may induce over-smoothing sooner, contributing to an overall degradation in performance. We investigate the relationship between the model's depth and performance. As depth is a hyper-parameter, we perform a search on 2, 4, 8, and 16 layers configurations — while maintaining identical searching parameters — on several datasets at an increasing rate of homophily. We then normalize our final accuracy results for each graph against the result of our 2-layer configuration to obtain relative performance gain in percentage. We illustrate our results in Figure(3).

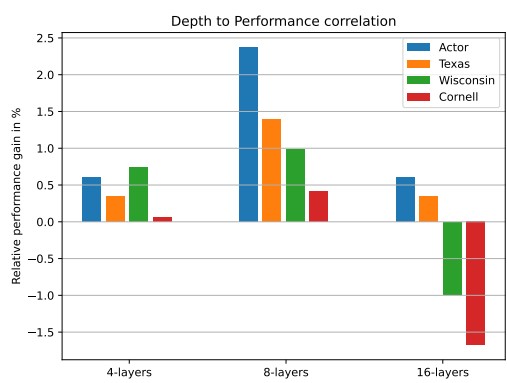

Figure 3: Illustration of relative performance against 2-layer configuration (not shown).

**Analyzing the augmented graphs**. We characterize the newly augmented graphs to understand better how they affect the overall performance by analyzing those searched under the eight-layers configuration. We first calculate the difference of the homophily rate between the new and old graphs. Next, we count the difference of total informative edges, i.e., edges between nodes of similar classes. Finally, we calculate the Intersection between edges of the original to the augmented graph. Table(3) summarizes our findings. Herein we observe that our improved graphs do not exhibit a stronger rate of homophily — in comparison to the original graphs — as initially assumed. However, we also observe that, despite the increasing heterophily, the searched graphs include more informative edges while retaining most of the original edges, indicating we are learning new and relevant unseen relationships. Nevertheless, this observation challenges the current research assumption on the correlation between heterophily and performance. We show that performance can still be achieved under low homophily given that enough informative edges are added to the graph and a deeper architecture.

**Analyzing the method's effectiveness and efficiency**. To evaluate the efficiency of our design, we compare its memory usage and total run time to other NAS-based approaches such as GraphNAS (Gao et al., 2019) and SANE (Zhao et al., 2021). Table(4) summarizes our findings. We could observe: that AutoCoG maintains relatively low memory utilization for each of the datasets tested, and AutoCoG is also the fastest model to complete both its full-search and training stages.

Table 3: We characterize the new graph structured found after search. From left-to-right, $\nabla H$ represents the change of homophily rate, $\nabla|\hat{E}|$ denotes the change in informative edges, and IoU describes the overlap between the original edges with the current edges

| dataset | $\nabla\mathcal{H}$ | $\nabla|\hat{E}|$ | Intersection |
|---|---|---|---|
| Actor | ↓ 0.056 | ↑ 4654 | 99.72% |
| Texas | ↑ 0.018 | ↑ 260 | 94.31% |
| Wisconsin | ↓ 0.059 | ↑ 263 | 96.93% |
| Cornell | ↓ 0.162 | ↑ 161 | 89.54% |

**Analyzing the effectiveness of progressive search**. To test the benefit of progressive search, we perform three ablation studies on Texas, Wisconsin, and Cornell at various depths. As shown in Table(5), note that, with progressive search, accuracy is positively correlated to a model's depth,

Table 4: We characterize the efficiency and effectiveness of our search method by measuring the memory usage and total run time to search and fully train a model for various datasets.

| | Actor | | Texas | | Wisconsin | | Cornell | |
|---|---|---|---|---|---|---|---|---|
| Model | GPU(MiB) | Run-time(s) | GPU(MiB) | Run-time(s) | GPU(MiB) | Run-time(s) | GPU(MiB) | Run-time(s) |
| SANE | 3890 | 3788 | 1070 | 2686 | 998 | 2194 | 994 | 3024 |
| GraphNAS | 1088 | 4320 | 1268 | 5161 | 1326 | 5262 | 972 | 4642 |
| **Auto-CoG** | 2634 | 960 | 992 | 427 | 994 | 450 | 984 | 360 |

while the opposite is true when searching without it. This is due to the search instability from the resulting search space size, and a deeper network only further exacerbates the problem. Indeed, we further observe that simply applying GDAS, as in the case of searching without progressive search, only yields poor architectures. The results align with our reasoning in section 3.2.2.

**Analyzing graph-model co-search improvement on model's robustness**. Table (6) summarizes our improved robustness on noisy graphs. We validate our robustness by performing node classification on graphs with added edges. The number of new edges corresponds to a percentage relative to the total edges. From the table, we make the following observations: First, noisy graph data leads to poor model performance from a lack of robustness. Second,

Table 5: We study the effectiveness of progressive search (PS) by comparing **Auto-CoG**'s performance at various depth search with and without it.

| | Texas | | | Wisconsin | | | Cornell | | |
|---|---|---|---|---|---|---|---|---|---|
| Mode | 2 | 4 | 8 | 2 | 4 | 8 | 2 | 4 | 8 |
| With PS | 77.30 | 77.57 | 80.27 | 79.96 | 80.20 | 80.39 | 64.86 | 64.32 | 64.59 |
| Without PS | 76.76 | 73.24 | 64.86 | 77.84 | 63.92 | 68.23 | 60.27 | 59.2 | 61.08 |

Graph-search can rectify and improve the model's robustness by removing artificial noise, leading to better model performance. Third, random edges can help improve some sparse graphs base performance, as in the case of Cornell and Wisconsin, by enabling skip-connections between distance neighborhoods.

Table 6: We analyze the improved robustness provided by graph-model co-search. First taking the original graphs, we added random noisy edges to the percentage amount, with respect to the total edges, specified in each column.

| | Texas | | | Wisconsin | | | Cornell | | |
|---|---|---|---|---|---|---|---|---|---|
| Mode | 20% | 40% | 80% | 20% | 40% | 80% | 20% | 40% | 80% |
| Model Only | 74.59 ± 1.39 | 73.78 ± 1.30 | 71.35 ± 2.28 | 65.88 ± 2.11 | 72.35 ± 0.62 | 71.35 ± 2.28 | 54.59 ± 1.13 | 61.24 ± 2.61 | 62.70 ± 1.13 |
| Co-Search | 80.81 ± 0.85 | 79.73 ± 1.91 | 77.29 ± 1.39 | 80.0 ± 1.24 | 81.56 ± 1.01 | 81.96 ± 0.83 | 66.22 ± 2.29 | 63.24 ± 1.88 | 65.40 ± 1.70 |

## 4.3 Results

We compare **Auto-CoG** to several notable state-of-the-methods inferencing on graphs with increasing degrees of associativity from 0.3 and 0.9. Additionally, we also include *average improvement* and *average rank* for quick performance comparison at a glance. *Average improvement* is the average accuracy difference between *Auto-CoG* and another model across all datasets, so a higher score indicates a better result. *Average rank* is a model's average performance rank for all datasets, so lower is better. For comparison, we include:

- NAS based graph models: for this category, we include GraphNAS(Gao et al., 2019), SANE(Zhao et al., 2021) and GASSO(Qin et al., 2021).

- Handcrafted graph models: we compare against traditional designs such as GCN (Kipf and Welling, 2016a), SGC (Wu et al., 2019b), GAT (Veličković et al., 2018), GCNII (Chen et al., 2020a),

Table 7: Test Accuracy (%) comparison with other previous state-of-the-art frameworks. Experiments are conducted on the WebKB, Coauthor, Amazon, and Actor datasets. To highlight only the model's performance, we select the best accuracy from each model among different depths between two to eight layers for each dataset. (*) best result. (**) second best result.

| Model | Actor | Texas | Wisconsin | Cornell | Computer | CS | Photos | Physics | Avg improv. | Avg Rank |
|---|---|---|---|---|---|---|---|---|---|---|
| SGC | 26.17±1.15 | 56.41±4.25 | 51.29±6.44 | 58.57±3.44 | 37.53±0.20 | 70.52±3.96 | 26.60±4.64 | 91.46±0.48 | ↑24.30 | 9.87 |
| GCN | 28.82±0.13 | 65.95±2.76 | 57.84±1.81 | 54.05±0.00 | 81.62±2.11** | 91.83±0.50** | 79.76±3.14 | 93.68±0.22 | ↑7.43 | 5.50 |
| GAT | 28.24±0.36 | 62.16±1.21 | 52.55±1.92 | 53.78±1.46 | 77.74±2.02 | 89.27±0.46 | 74.56±3.02 | 93.19±0.36 | ↑10.18 | 8.13 |
| GCNII | 34.28±1.12** | 69.19±6.56 | 70.31±4.75 | 61.08±2.76 | 37.56±0.43 | 71.67±2.68 | 62.95±9.41 | 93.15±0.92 | ↑14.10 | 5.75 |
| JKNet | 28.80±0.97 | 61.08±6.23 | 52.76±5.69 | 57.30±4.95 | 67.99±5.07 | 81.82±3.32 | 78.42±6.95 | 90.92±1.61 | ↑11.73 | 8.00 |
| APPNP | 28.65±1.28 | 60.68±4.50 | 54.24±5.94 | 58.43±3.74 | 43.02±10.16 | 91.61±0.49 | 59.62±23.27 | 93.75±0.61** | ↑15.74 | 7.13 |
| Geom-GCN | 31.63±0.02 | 65.94±1.39 | 68.63±0.00 | 59.75±1.80 | — | — | — | — | ↑9.40 | 5.5 |
| H2GCN | 33.13±0.10 | 82.41±0.07* | 79.61±1.01** | 80.4±0.05* | 37.48±0.08 | 28.83±7.95 | 46.56±0.17 | 93.90±0.05* | ↑16.33 | 4.75 |
| GraphNAS | 26.87±2.09 | 78.11±3.91 | 63.14±5.13 | 59.73±4.49 | 84.66±0.22* | 90.11±0.31 | 91.11±0.18* | 93.75±0.60 | ↑3.18 | 4.25 |
| SANE | 32.05±1.49 | 71.89±7.77 | 60.39±10.57 | 54.59±11.02 | 78.99±4.3 | 88.51±0.65 | 87.72±1.50 | OOM | ↑6.50 | 5.20 |
| GASSO | 27.02±0.05 | 64.86±0.00 | 78.43±0.00 | 64.70±0.00 | OOM | OOM | 89.32±0.05** | OOM | ↑4.88 | 5.00 |
| **Auto-CoG** | 38.04±0.16* | 80.27±2.21** | 80.39±0.00* | 64.86±2.97** | 78.91±2.57 | 92.05±0.40* | 85.16±1.12 | 93.28±0.58 | – | 2.50 |

JKNet (Xu et al., 2018) and APPNP (Klicpera et al., 2018). Additionally, we also compare against designs that are crafted specifically for disassortative graphs such as Geom-GCN (Pei et al., 2020c) and H2GCN (Zhu et al., 2020b).

We summarizes our finding in Table(7). From the results, we make the following observations:

- Highlighting the challenge of heterophily, we observe the lack of a dominant approach that can outperform all datasets. However, when we compare their average ranking overall, we do find **Auto-CoG** ranks highly at 2.5 and able to improve against all other approaches on average. This showcase our method's robustness in dealing with graphs under different homophily settings

- In comparison to other NAS approaches, **Auto-CoG** reliably outperforms all of them when it comes to disassociative datasets since typical GNNs tend to over-smooth on noisy graph data — an inherent problem for message-passing. **Auto-CoG** directly modifies its graph data and network's architecture to overcome this weakness. GASSO (Qin et al., 2021) also performs graph structure search, but it is limited to only learning existing edges attention coefficients and therefore is still susceptible to some degree of over-smoothing.

- In comparison to handcrafted baselines, **Auto-CoG** comfortably outperforms Geom-GCN (Pei et al., 2020c) on Actor and WebKB datasets. Our graph-structured learning process provides a similar function as the "structural neighborhood" concept, which Geom-GCN utilizes for bi-level aggregation. On the other hand, H2GCN (Pei et al., 2020c) shows impressive performance on small Webkb datasets, outperforming *Auto-CoG* in both Texas and Cornell. However, the model's 'ego-embeddings' concept does not scale well on larger datasets such as CoAuthor and Amazon, where it repeatedly fails to produce competitive results.

## 5 Conclusion and Limitations

In this paper, we present **AutoCoG** the first NAS framework towards **unified data-model co-search for GNNs**. Our results convincingly demonstrate the benefit of data-graph co-search for both deep and shallow graph neural networks. Our ablation study shows that controlled variances in graph heterophily can result in a better, more generalized model and the necessity for graph-augmentation to be model-aware. We confidently demonstrate **AutoCoG** to be a reliable way to discover robust architectures, a stable model training environment, and state-of-the-art results. Additionally, we show that the localized disturbance of graph structure motivates node position learning, allowing for greater generalizability of the model.

However, there are still limitations that need to be addressed: large graph scalability and understanding heterophily's relationship to performance. To address this, we first plan to follow

up by learning meaningful model/graph using **Auto-CoG** via graph-batching. Secondly, we want to conduct a study to understand better the phenomenon between heterophily and performance observed in our ablation. There is no negative societal impact to our best knowledge, except that the NAS process is resource-consuming - but even that excessive cost can be amortized by the re-usability of the searched model, which can achieve superior accuracy-resource trade-off.

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

## A Appendix

### A.1 Datasets

We evaluate **AutoCoG** on several popular semi-node classification datasets including (i) the WebKB datasets —Cornell, Texas, Wisconsin— (Craven et al., 1998), (ii) Actor dataset (Tang et al., 2009), (iii) Co-author datasets —CS, Physics— (Shchur et al., 2019), and (iv) Amazon datasets — Photo, Computers — (Shchur et al., 2019). We select these datasets to represent a wide range of graphs under different degrees of homophily, which will serve to demonstrate **Auto-CoG** robustness in comparison to the different SOTA methods. The specifics on each datasets, in sorted order under homophily, is recorded in Table(8).

Table 8: The Statistics of each dataset. From left to right: unique classes, nodes, edges and embedding dimension and edge-homophily degree.

| dataset | $|Y|$ | $|V|$ | $|E|$ | $|D|$ | $\mathcal{H}$ |
|---|---|---|---|---|---|
| Actor | 5 | 7,600 | 33,544 | 931 | 0.375 |
| Texas | 5 | 183 | 295 | 1,703 | 0.411 |
| Wisconsin | 5 | 251 | 499 | 1703 | 0.488 |
| Cornell | 5 | 183 | 295 | 1,703 | 0.567 |
| Computers | 10 | 13,752 | 491,722 | 767 | 0.783 |
| CS | 40 | 18,333 | 163,788 | 6,805 | 0.827 |
| Photos | 10 | 7,650 | 238,162 | 745 | 0.833 |
| Physics | 5 | 34,493 | 495,924 | 8,415 | 0.936 |

---

**Algorithm 1: AutoCoG** Searching Algorithm

---

**Input:** $W_s$, $X$, $G(V, E)$, searchSpace, epochs, startNumLayer, endNumLayer, stages ;
**Output:** $\alpha$, $G(V, \bar{E})$, $S$;
$\bar{E} \leftarrow E$ ;
**for** *s = 0 to stages-1* **do**
    #Initialize new model and architecture parameters
    $\alpha \leftarrow$ *OnesInitParameters*(searchSpace);
    $W \leftarrow$ *initModel*(*min*(startNumLayer+s, endNumLayer));
    **for** *e=0 to epochs-1* **do**
        $S \leftarrow \sigma(\text{MLP}(f(X, G(V, \bar{E}); W_s))$;
        $\bar{a} \leftarrow$ *Sample*($\alpha$);
        BackPropgate $L_{obj}(\bar{a}, W, S, X_{train}, G(V, \bar{E})) \rightarrow W, W_s$;
        BackPropgate $L_{obj}(\bar{a}, W, S, X_{valid}, G(V, \bar{E})) \rightarrow \alpha$;
    **end**
    #Reduce search space and augment edges
    $\bar{E} \leftarrow$ *PruneAndGrow*($f(.; W_s)$);
    searchSpace $\leftarrow$ *ReduceSearchSpace*(searchSpace, $\alpha$);
**end**
$S \leftarrow \sigma(\text{MLP}(f(X, G(V, \bar{E}); W_s))$;

---

