# AutoCoG: A Unified Data-Model Co-Search Framework for Graph Neural Networks

**Anonymous**[1]

[1]Anonymous Institution

**Abstract** Neural architecture search (NAS) has demonstrated success in discovering promising architectures for vision or language modeling tasks, and it has recently been introduced to searching for graph neural networks (GNNs) as well. Despite the preliminary success, GNNs struggle in dealing with heterophily or low-homophily graphs where connected nodes may have different class labels and dissimilar features. To this end, we propose co-optimizing both the input graph topology and the model's architecture topology simultaneously. That yields **AutoCoG**, the first unified data-model co-search NAS framework for GNNs. By defining a highly flexible data-model co-search space, **AutoCoG** is gracefully formulated as a principled bi-level optimization that can be end-to-end solved by the differentiable search methods. Experiments show **AutoCoG** achieves an average performance gain across all datasets of 3.18% over the following best approach and ranks best against all other state-of-the-art methods with an average ranking of 2.5.

## 1 Introduction

Graph neural networks (GNNs) have emerged as promising tools to analyze networked data in various real-world scenarios, such as social media Grover and Leskovec (2016) and biochemical graph analytics Zitnik and Leskovec (2017). Specifically, GNNs apply recursive message passing to learn the embedding representation of each node via aggregating the representations of its neighbors and itself. Motivated by the significant success of node embedding learning, plenty of GNN variants have been explored for the diverse downstream graph analysis tasks, including GCN Kipf and Welling (2016a), GraphSAGE Hamilton et al. (2018), and GCNII Chen et al. (2020a).

However, training GNNs is notoriously challenging, more so when they are train under heterophily or disassociative graphs, not to mention deep GNNs Chen et al. (2020a). First, since graphs abstract diverse data sources and present tremendous heterogeneity, the success of GNNs is often accompanied by extensive tuning of model architectural hyperparameters to characterize specific graph data. For example, it was reported that graph attention networks GAT Veličković et al. (2018) are sensitive to the number of attention heads, which has to be carefully searched for the citation networks and the protein-protein interaction data, respectively. Second, in the real world graphs often opposites attract which inevitably lead to noisy setting where GNNs tend to suffer from overfitting and generalize poorly to the unseen testing data. Third, despite the potential of deep GNNs in learning the informative high-order neighborhood, the training of deep GNNs is widely known to be limited by the issues of over-smoothing, gradient vanishing, and over-squashing Chen et al. (2020a).

Recently, the automated graph neural architecture search (NAS), graph augmentation tricks, and deeper architectures have been independently proposed to tackle the above GNN training challenges partially. Expressly, most of the existing automated efforts are limited to neural architecture tuning, while graph augmentation is often overlooked and untouched despite often being effective to gain performance Li and King (2020); Zhou et al. (2019a). This is primarily because changes to the existing graph structure can have a cascading effect on the process of information aggregation, which adds a new layer of complexity above the already complex architecture tuning problem.

Submitted to AutoML Conference 2022

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

$$\frac{|\{(v, w) : (v, w) \in E \wedge y_v = y_w\}|}{E} \tag{1}$$

Second, given a GCN, its $k$-th layer could be written generally as:

$$
\begin{aligned}
h_i^{(k)} &= \text{AGGR}(\{a_{ij}^{(k)} W^{(k)} x_j^{(k-1)} : j \in \mathcal{N}(i)\}) \\
x_i^{(k)} &= \sigma(\text{COMB}(W^{(k)} x_i^{(k-1)}, h_i^{(k)}))
\end{aligned}
\tag{2}
$$

$x_i^{(k)}$ denotes the node embedding of element $i$ at $k$-th layer. $W^{(k)} \in \mathbb{R}^{D \times D}$ represents the learnable layer-wise weights for all $\{x_i : i \in |V|\}$, where $|V|$ is our total number of nodes and $D$ our number of hidden features. $a_{ij}^{(k)}$ dictates the attention coefficient between $i$ and $j$ derived from some Attention function. $\mathcal{N}(i)$ denotes the neighboring nodes of node $i$ from a graph $G$. $h_i^{(k)}$ is the resulting embeddings after applying an AGGR function to aggregate a set of neighboring embeddings from the previous $k-1$ layers. In addition, function COMB incorporates information from

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

---