# OpenReview forum: "AutoCoG: A Unified Data-Model Co-Search Framework for Graph Neural Networks"
_automl.cc/AutoML/2022/Track/Main — AutoML-Conf 2022 (Main Track)_

### Official Review · Reviewer_Y6UV · 2022-03-30

**Potential Impact On The Field Of Automl Rating:** 3
**Technical Quality And Correctness Rating:** 2
**Clarity Rating:** 3

**Summary Of Contributions:**

This paper proposes a joint framework to search both model architecture and graph structure.
The searching problem setup is well defined.
To solve the problem of directly applying GDAS to GNN search, the progressive search space is introduced to ease the search difficulty and identity mapping is introduced to relieve the over-smoothing and over-fitting issues.
The experimental analysis is comprehensive and the results are significant.

**Clarity:**

The presentation in this paper is smooth and easy to follow. But there are several minor issues.

line 25, trained; line 41, because of; line 79, analysis on; line 99, have; inconsistent of over-xxx and overxxx.



**Overall Review:**

Positive:
1. Novelty. This paper proposes a joint framework to search both model architecture and graph structure, which is new to the literature.
2. Technically sound. The main components, including the model search space, graph augmentation mechanisms, optimization, bi-level optimization and some stabilize tricks are sound and well described.
3. Experiment results are analysised in different aspects.

Negative:
1. Lack of efficiency evaluation. As known, efficiency is an important problem in AutoML. Since this paper searches for both model and data, the effiiency issue should be seriously considered.
2. Lack of comparison with AutoML baselines. In section 2, the authors mentioned several NAS for GNN approaches. But it seems that the main comparison in experiments does not include these methods.
3. Code is not provided and running environment is not clear.


**Potential Impact On The Field Of Automl:**

This paper can bring some insight with two aspects:
(1) The joint searching/optimization of both architecture and data augmentation.
(2) The new AutoML approach for graph learning.


**Reproducibility:**

There lacks code in this paper. The running environment is also missing.

**Review Confidence:**

3: You are fairly confident in your assessment. It is possible that you did not understand some parts of the submission or that you are unfamiliar with some pieces of related work.

**Review Rating:**

4: Marginally above the acceptance threshold (use sparsely)

**Review Summary:**

Overall, this paper brings something new to the AutoML and GNN literature. The paper is sound and well-written. But additional experiment is needed. So I recommend a revision to add some new experimental comparison in terms of effieincy and baselines.

**Technical Quality And Correctness:**

The main components, including the model search space, graph augmentation mechanisms, optimization, bi-level optimization and some stabilize tricks are sound and well described.
The experiment setup is clear. The designed mothed is evaluated with several ablation studies. And the results look significant compared with several SOTA gnn methods. However, the search efficiency is not evaluated and there lacks the comparison with other AutoML methods for GNN, such as GraphNAS (Gao et. al.) and SNAG (Zhao et. al.).

---

### Official Review · Reviewer_SWXw · 2022-04-04

**Potential Impact On The Field Of Automl Rating:** 3
**Technical Quality And Correctness Rating:** 3
**Clarity Rating:** 3

**Summary Of Contributions:**

The authors propose a neural architecture search framework with data model co-search for GNN, especially under heterophily conditions. The neural model search and input graph topology is optimized simultaneuously. Towards this the authors design a bi-level optimization framework. The idea  is interesting and important especially for graphs under heterophily.
The authors compare their work on different datasets with different degree of homophily
The experiments establish superior performance of the proposed architecture.



**Clarity:**

The paper is very well written. Definitions of homophily etc are clearly defined. The motivation is crisp.

**Ethics Details (Optional):**



**Overall Review:**

Overall the work is good. The problem tackled is interesting. The method is innovative for the task in hand. Experiments are performed on a wide number of datasets under different setting. There is a small doubt on reproducibility due to non-availability of code and clarity on parameters of baselines.

Also, I am not very confident on the baselines used? Are there works that tackle heterophily specifically, that need to be compared with?

**Potential Impact On The Field Of Automl:**

The idea is interesting and can help in improving GNNs performance by finding effective model parameters. This is interesting, especially in case of graphs under heterophily which are challenging to model.

**Reproducibility:**

Code is not available as of now, however the authors have declared to provide code upon acceptance.

 The parameters of the baseline methods are not clear to me. Were they optimized for the task in hand? This particular part is not clear to me. Were they optimized only for depth? Is this sufficient(and also reasonable)?

Also, I am not very confident on the baselines used? Are there works that tackle heterophily specifically, that need to be compared with?

**Review Confidence:**

3: You are fairly confident in your assessment. It is possible that you did not understand some parts of the submission or that you are unfamiliar with some pieces of related work.

**Review Rating:**

5: Accept, good paper

**Review Summary:**

Overall, the paper is good. Just need some clarity on the setup of baselines(their parameters etc.).

**Technical Quality And Correctness:**

The technical contribution to solve this problem is interesting.  The experiments are designed to compare with different settings of heterophily. Further, ablation studies are included to justify importance of components.

---

### Official Review · Reviewer_2Lu1 · 2022-04-04

**Potential Impact On The Field Of Automl Rating:** 2
**Technical Quality And Correctness Rating:** 2
**Clarity:** The presentation of the work is clear.
**Clarity Rating:** 3

**Summary Of Contributions:**

This work proposes a differentiable data-model co-search framework, AutoCoG,  for GNNs, which formulates the problem as a bi-level optimization process, and enables the end-to-end discovery of graph structure and GNN architecture. The experiment results validate the effectiveness of AutoCoG to search for deep networks for graphs under either homophily or heterophily.

**Overall Review:**

### Strengths:

1. The presentation of the paper is clear.

2. The experimental results of the proposed framework on datasets under both homophily and heterophily is better than or competitive with the baselines, which demonstrates its effectiveness.

3. The analysis with regard to the network depth and the augmented graph provides interesting insights.

### Weaknesses:
1. The major concern for this paper is its limited novelty. The idea of data-model co-search for GNNs has already been proposed by a previous work [1], with the similar motivation, problem formulation and approach as AutoCoG. However, there is no discussion with this piece of important related work.

2. The experiments are not extensive enough to support the authors’ claim. First, the graph NAS baselines, e.g., [2][3], and the GNN designed especially for graph heterophily,e.g., [4][5] are missing. Besides performance gain, the necessity of searching for graph augmentation, e.g., increasing the robustness to noisy graph structure, lacks enough validation. And the effectiveness of the tricks in 3.2.2 is not validated by the corresponding ablation study.

3. There lacks the theoretical or empirical analysis with regard to the model complexity and the search efficiency.

## Refs:

[1] Graph Differentiable Architecture Search with Structure Learning. NeurIPS 2021.

[2] Graphnas: Graph neural architecture search with reinforcement learning. IJCAI 2019.

[3] Search to Aggregate Neighborhood for Graph Neural Network. ICDE 2021.

[4] Geom-gcn: Geometric graph convolutional networks. ICLR 2020.

[5] Generalizing Graph Neural Networks Beyond Homophily. NeurIPS 2020.


**Potential Impact On The Field Of Automl:**

This paper provides an interesting perspective for automatically designing high-performing GNN architectures: searching for better graph structure and model, simultaneously. However, given that there already existed another paper [1] with the similar motivation and method, I’m afraid the impact of this submitted work on the field of AutoML will be limited.

[1] Graph Differentiable Architecture Search with Structure Learning. NeurIPS 2021.


**Reproducibility:**

The reproducibility list is filled out in a reasonable way and the reproducibility of the work is believed to be high.

**Review Confidence:**

5: You are absolutely certain about your assessment. You are very familiar with the related work and checked all the details carefully.

**Review Rating:**

4: Marginally above the acceptance threshold (use sparsely)

**Review Summary:**

Considering the limited novelty and the unsolid experiments, I recommend a rejection of the paper.

**Technical Quality And Correctness:**

The proposed approach is technically sound. However, some of the authors’ claims are not supported by either theories or experiments.

---

### Meta-Review · Area_Chair_KRWx · 2022-05-09

**Recommendation:** Accept
**Confidence:** 4

**Metareview:**

The authors propose a bi-level end-to-end optimization process that encompasses both discovery of graph structure and GNN architecture. Initial concerns regarding closely related work, missing baselines, and lack of efficiency evaluation have been addressed. No major concerns are left and all reviewers recommend acceptance due to the work's novelty and empirical rigour. I agree with their assessment and am convinced that the provided work would contribute to the diversity of problems presented at AutoML-Conf.

---

### Decision · Program_Chairs · 2022-05-13

Accept